# Plasma Copper Concentration Is Associated with Cardiovascular Mortality in Male Kidney Transplant Recipients

**DOI:** 10.3390/antiox12020454

**Published:** 2023-02-10

**Authors:** Manuela Yepes-Calderón, Daan Kremer, Adrian Post, Camilo G. Sotomayor, Ulrike Seidel, Patricia Huebbe, Tim J. Knobbe, Kai Lüersen, Michele F. Eisenga, Eva Corpeleijn, Martin H. De Borst, Gerjan J. Navis, Gerald Rimbach, Stephan J. L. Bakker

**Affiliations:** 1Department of Internal Medicine, Division of Nephrology, University Medical Center Groningen, 9713 GZ Groningen, The Netherlands; 2Clinical Hospital University of Chile, University of Chile, Independencia 8380453, Chile; 3Institute of Human Nutrition and Food Science, University of Kiel, 24118 Kiel, Germany; 4Department of Epidemiology, University Medical Center Groningen, 9713 GZ Groningen, The Netherlands

**Keywords:** copper, cardiovascular, mortality, kidney transplantation

## Abstract

Kidney transplant recipients (KTR) are at increased risk of cardiovascular mortality. We investigated whether, in KTR, post-transplantation copper status is associated with the risk of cardiovascular mortality and potential effect modification by sex. In this cohort study, plasma copper was measured using mass spectrometry in extensively-phenotyped KTR with a functioning allograft >1-year. Cox regression analyses with the inclusion of multiplicative interaction terms were performed. In 660 KTR (53 ± 13 years old, 56% male), the median baseline plasma copper was 15.42 (IQR 13.53–17.63) µmol/L. During a median follow-up of 5 years, 141 KTR died, 53 (38%) due to cardiovascular causes. Higher plasma copper was associated with an increased risk of cardiovascular mortality in the overall KTR population (HR 1.37; 95% CI, 1.07–1.77 per 1-SD, *p* = 0.01). Sex was a significant effect modifier of this association (P_interaction_ = 0.01). Among male KTR, higher plasma copper concentration was independently associated with a two-fold higher risk of cardiovascular mortality (HR 2.09; 95% CI, 1.42–3.07 per 1-SD, *p* < 0.001). Among female KTR, this association was absent. This evidence offers a rationale for considering a sex-specific assessment of copper’s role in cardiovascular risk evaluation. Further studies are warranted to elucidate whether copper-targeted interventions may decrease cardiovascular mortality in male KTR.

## 1. Introduction

Kidney transplantation is the best available treatment for end-stage kidney disease [1]. Great improvements have been made in the short-term survival of kidney transplant recipients (KTR); however, achieving similar results in long-term survival remains an ongoing challenge [2]. Death with a functioning graft accounts for half of the long-term graft-loss cases [2,3], and cardiovascular mortality plays a leading role among the causes of premature death in KTR [4]. Above being subject to traditional cardiovascular risk factors, KTR are also subject to complex pathophysiological processes, including enhanced oxidative stress, pro-inflammatory status, and trace elements imbalance [5,6,7].

Copper is a trace element that has gained attention as an enhancer of cardiovascular risk [8,9,10,11,12]. The amount of copper ingested with food and water is relatively low [13], mainly determined by its presence in the soil [14]. Under normal conditions, plasma copper concentration is tightly regulated, and the organism can account for excess copper intake [15]. However, increased plasma copper concentration has been demonstrated in patients with end-stage kidney disease and the early stages after kidney transplantation [16,17]. In addition, mycophenolate therapies can further increase plasma copper concentration in outpatient KTR [5,18].

Multiple mechanisms link copper with cardiovascular disease, including promoting LDL oxidation to its atherogenic state [19,20,21], promoting central inflammation [22,23], and participating in the formation of reactive oxygen species [12,13,24]. In the general population, cross-sectional studies have shown higher plasma copper concentration in patients with coronary artery disease and heart failure compared to healthy controls [25,26,27]. Moreover, multiple longitudinal studies have connected higher plasma copper concentration with an increased risk of stroke, acute myocardial infarction, and cardiovascular mortality [8,9,10,11]. Notably, estrogen is known to have a mitigating effect on copper-mediated LDL oxidation [28]. However, studies in the general population were characterized by a preponderance of male population which precluded investigating a potential sex-specific association of copper with cardiovascular risk [11].

To date, no study has investigated the association between post-transplantation plasma copper concentration and the risk of cardiovascular mortality in KTR. In the current study, we primarily test the hypothesis that plasma copper concentration is associated with the long-term risk of cardiovascular mortality in KTR. Next, we explore potential differences among male and female KTR.

## 2. Materials and Methods

### 2.1. Study Design

In this cohort study, all adult KTR with a functioning graft for longer than one year, who visited the outpatient clinic of the University Medical Center of Groningen (The Netherlands) between November 2008 and May 2011, were invited to participate. Patients with a history of addiction or malignancy were excluded. Seven hundred and seven eligible KTR signed written informed consent. The information about 660 KTR with available plasma copper measurement at baseline is presented here. This study was approved by the Institutional Review Board (METc 2008/186) and adhered to the Declarations of Helsinki and Istanbul.

### 2.2. Data Collection

Baseline clinical data were collected during a visit to the outpatient clinic, following a detailed protocol extensively described elsewhere [29]. Other relevant transplant, donor, and recipient data were extracted from the Groningen Kidney Transplant Database. This database holds information on all kidney transplants performed at our center since 1968 [30]. Cause of death was obtained by linking the number of the death certificate to the primary cause of death as coded by a physician from the Central Bureau of Statistics. Endpoints were recorded until September 2015, resulting in a median follow-up of 5.4 (IQR 4.8–6.1) years after inclusion. We contacted general practitioners or referring nephrologists in cases where the status of a patient was unknown. No participants were lost to follow-up.

### 2.3. Laboratory Measurements, Calculations, and Definitions

During the first medical visit after study inclusion, at a median of 5.2 (IQR 1.8–12.1) years after transplantation, all participants provided fasting blood samples. Aliquots of plasma generated from these samples were stored frozen at −80 °C and plasma copper was afterwards measured using inductively coupled plasma mass spectrometry (ICPMS) ICAP Q instrument (Thermo Fisher Scientific, Waltham, MA, USA) at Synalb (Jena, Germany). Measurements were conducted following DIN EN ISO 17294-2: 2017-01 [31]. This test has a detection limit of 0.1 µg/L, an intra-assay variability of 0.3%, and inter-assay variability of 1%.

LDL cholesterol was calculated using the Friedewald equation. Estimated glomerular filtration rate (eGFR) was calculated using the serum creatinine-based Chronic Kidney Disease EPIdemiology collaboration equation (CKD-EPI). Cardiovascular mortality was defined as the principal cause of death being cardiovascular in nature (codes 410–447 of the International Statistical Classification of Diseases-9) [32]. Proteinuria was defined as urinary protein excretion ≥ 0.5 g/24 h.

### 2.4. Statistical Analyses

Data analyses, computations, and graphs were performed with R version 4.0.5 (R Foundation for Statistical Computing, Vienna, Austria). The distribution of quantitative variables was assessed using Quantile-Quantile plots. Data are presented as mean ± standard deviation [SD] for normally distributed data and as median (interquartile range [IQR]) for variables with a non-normal distribution. Categorical data are expressed as numbers (percentages). Differences in baseline characteristics among subgroups of KTR according to tertiles of plasma copper concentration were tested using one-way ANOVA for continuous variables with normal distribution, Mann–Witney U test for continuous variables with non-normal distribution, and χ^2^ test for categorical variables.

The risk of cardiovascular mortality was tested using crude and multivariable Cox proportional-hazards regression analyses. Initial adjustment was performed for age, sex, and body mass index (BMI) in model 1. Successive regression models were built in a forward stepwise fashion. Further adjustment was performed for eGFR, time since transplantation and dialysis time before transplantation in model 2; calcineurin inhibitor use and proliferation inhibitor use in model 3; systolic blood pressure (SBP), history of previous cardiovascular disease and hemoglobin in model 4; total cholesterol, LDL cholesterol, and diabetes history in model 5, and Short QUestionnaire to ASsess Health–enhancing physical activity (SQUASH) score and high-sensitivity C-reactive protein (hs-CRP) in model 7.

Next, we performed pre-specified effect-modification analyses by sex and other variables associated with copper status or cardiovascular risk in KTR [3,15]. To do so, we used multiplicative interaction terms. If a significant interaction was proven, we proceeded with stratified longitudinal analyses. First, to visualize the continuous associations of plasma copper concentration with cardiovascular mortality in each subgroup, plasma copper concentration, as a continuous variable, was plotted against the risk of cardiovascular mortality. Next, multivariable Cox proportional-hazards regression analyses were performed, following the adjustment models previously described.

#### Secondary Analyses

As secondary analyses, we performed linear regression analyses to explore the association of plasma copper concentration with variables related to its proposed pathophysiology in cardiovascular disease (total cholesterol, LDL cholesterol, and hs-CRP) [19,20,21,22,23], both in the overall KTR population and among subgroups. We also performed competing-risk analyses using Cox proportional-hazards regression analyses to test the association of plasma copper concentration with the competing outcomes of all-cause and non-cardiovascular mortality [33], and tested the association of plasma copper concentration with the risk of graft failure. Finally, as sensitivity analyses, we evaluated the association of plasma copper concentration with the risk of cardiovascular mortality with the exclusion of KTR (i) with eGFR < 30 mL/min/1.73 m^2^, (ii) who died within the first year of follow-up, (iii) and KTR outside the −2 and +2 SD of plasma copper concentration. Since the number of events was reduced, these analyses were only adjusted according to Model 1.

For all regression analyses, plasma copper concentration was transformed to its natural logarithm and later standardized. Results are presented as hazard ratios (HR) per 1 standard deviation (SD) increase of natural log-transformed plasma copper concentration. A statistical significance level of *p* < 0.05 (two-tailed) was used for all statistical analyses, and we checked the models for the fulfillment of the assumptions of linear and Cox regression.

## 3. Results

### 3.1. Baseline Characteristics

A total of 660 outpatient KTR were included (mean age 53 ± 13-year-old, 56% male, mean eGFR 52 ± 20 mL/min/1.73 m^2^) at a median of 5.2 (IQR 1.8–12.1) years after transplantation. Baseline characteristics of the overall KTR population, and by tertiles of plasma copper concentration, are shown in Table 1. Median plasma copper was 15.42 (IQR 13.53–17.63) µmol/L. KTR in the highest tertile of plasma copper concentration were less often male (*p* < 0.001), had higher BMI (*p* = 0.01), lower SBP (*p* = 0.02), higher total and LDL cholesterol (*p* < 0.001 and *p* = 0.001; respectively), more often diabetes (*p* = 0.003) and lower hemoglobin concentration (*p* < 0.001). KTR in the highest tertile also had higher hs-CRP (*p* < 0.001) and were more physically inactive according to the SQUASH score (*p* = 0.001).

Plasma copper concentration was significantly lower among male KTR when compared to female KTR (14.64 [12.75–16.37] vs. 16.68 [14.64–19.20] µmol/L, respectively, *p* < 0.001; Figure 1). Therefore, we also tested the difference in baseline characteristics among sex-specific plasma copper concentration tertiles (Appendix A). Among male KTR, those in the highest tertile of plasma copper concentration were older (*p* = 0.002), had higher BMI and waist circumference (*p* = 0.01 and *p* = 0.001; respectively), had a higher prevalence of cardiovascular disease (*p* = 0.03) and higher total and LDL cholesterol (*p* = 0.004 and *p* = 0.001; respectively). Male KTR in the highest tertile of plasma copper concentration also had more often diabetes (*p* = 0.01), higher hs-CRP (*p* < 0.001), and a lower SQUASH score (*p* = 0.007). Among female KTR, those in the highest tertile of plasma copper concentration had a higher leucocyte count (*p* = 0.01) and hs-CRP (*p* < 0.001) without other significant differences.

### 3.2. Longitudinal Analyses

During a median follow-up of 5.4 (IQR 4.8–6.1) years, 141 (21%) KTR died, 54 (38%) due to cardiovascular causes. In the overall KTR population, plasma copper concentration was associated with a higher risk of cardiovascular mortality (HR 1.37, 95% CI 1.07–1.77 per 1-SD increment, *p* = 0.01). This association persisted upon adjustment for age, sex, BMI, and the variables included in models 2 to 5. However, it was not independent of adjustment for SQUASH score and hs-CRP (model 6, Table 2).

In pre-specified effect-modification analyses, we found that the association between plasma copper concentration and cardiovascular mortality was significantly modified by sex (P_interaction_ = 0.01). BMI, SPB, total and specific cholesterol, hs-CRP, and leucocyte count were not significant effect modifiers (Appendix A). We then performed stratified analyses by subgroups of KTR according to sex. Visualizing plasma copper concentration against the risk of cardiovascular mortality by sex showed an association between copper plasma concentration and cardiovascular mortality in male KTR. This association was absent for female KTR (Figure 2).

Among male KTR, a 1-SD higher plasma copper concentration was associated with a two-fold higher risk of cardiovascular mortality (HR 2.09, 95% CI 1.42–3.07 per 1-SD increment, *p* < 0.001). This association was independent of adjustment for all the variables included in the previously described models, with, for example, an HR of 1.80 (95% CI 1.21–2.69 per 1-SD increment, *p* = 0.004) after adjustment for age, BMI, total cholesterol, LDL cholesterol and history of diabetes (Table 3).

### 3.3. Secondary Analyses

We evaluated the association of known cardiovascular risk factors previously proposed to link copper with cardiovascular disease, both among the overall KTR population and among sex-specific subgroups. In the general KTR population, plasma copper concentration was positively associated with total cholesterol (Std. β = 0.14, *p* < 0.001), LDL cholesterol (Std. β = 0.13, *p* < 0.001), and hs-CRP (Std. β = 0.55, *p* < 0.001). Among male KTR, plasma copper concentration was also positively associated with total cholesterol (Std. β = 0.16, *p* < 0.001), LDL cholesterol (Std. β = 0.19, *p* < 0.001), and hs-CRP (Std. β = 0.57, *p* < 0.001). Among female KTR, plasma copper concentration was only associated with hs-CRP (Std.β = 0.58, *p* < 0.001), no association was found with total cholesterol (Std. β = 0.02, *p* = 0.72) or LDL cholesterol (Std.β = 0.04, *p* = 0.55, Figure 3).

We also performed competing risk analyses. In the overall KTR population, plasma copper concentration was not associated with the risk of all-cause or non-cardiovascular mortality (HR 1.12, 95% CI 0.97–1.30 per 1-SD increment, *p* = 0.13 and HR 1.02, 95% CI 0.83–1.26 per 1-SD increment, *p* = 0.84; respectively). In male KTR, there was an association between plasma copper concentration and all-cause mortality (HR 1.43, 95% CI 1.09–1.89 per 1-SD increment, *p* = 0.01). Still, it was not independent of adjustment by age and BMI, nor by adding the other variables included in the previously described models. In this same group, there was no association between plasma copper concentration and non-cardiovascular mortality (HR 1.03, 95% CI 0.72–1.47 per 1-SD increment, *p* = 0.88). In female KTR, plasma copper concentration was not associated with the risk of all-cause or non-cardiovascular mortality (HR 1.02, 95% CI 0.82–1.29 per 1-SD increment, *p* = 0.83 and HR 1.01, 95% CI 0.76–1.35 per 1-SD increment, *p* = 0.92; respectively). Regarding graft failure, there was no significant association between plasma copper concentration and the risk of graft failure in the overall and male KTR (HR 0.85, 95% CI 0.68–1.07 per 1-SD increment, *p* = 0.16 and HR 0.86, 95% CI 0.62–1.18 per 1-SD increment, *p* = 0.35; respectively), nor in female KTR (Appendix A).

### 3.4. Sensitivity Analyses

In sensitivity analyses, plasma copper concentration remained associated with the risk of cardiovascular mortality in male KTR in analyses performed after exclusion of KTR (i) with eGFR < 30 mL/min/1.73 m^2^ (HR 2.24, 95% CI 1.42–3.55 per 1-SD increment, *p* = 0.001), (ii) who died within the first year of follow-up (HR 1.74, 95% CI 1.10–2.75 per 1-SD increment, *p* = 0.02), and (iii) KTR outside the −2 to +2 SD of plasma copper concentration (HR 1.83, 95% CI 1.19–2.81 per 1-SD increment, *p* = 0.01; Appendix A).

## 4. Discussion

In a large cohort of outpatient KTR, plasma copper concentration was positively associated with cardiovascular mortality, with a significant effect modification by sex. Among male KTR, 1-SD higher plasma copper concentration was independently associated with a two-fold higher risk of cardiovascular mortality, while this association was absent among female KTR. These findings agree with previous studies linking copper status with cardiovascular disease in the general population [8,9,10,11,13,19,22,23,24,25,26,27]. To our best knowledge, this is the first study that explored this topic in the post-kidney transplantation setting.

Copper is an essential trace element usually present in small amounts in drinking water and food. Common sources of copper include shellfish, seeds, nuts, and organ meats [34]. Although the plasma copper concentration is tightly regulated in the general population [15,35], it is increased in both male and female KTR [16], making this population potentially more vulnerable to the harmful effects of copper overload. Potential explanations for the enhanced copper status of KTR include: (i) a remnant effect from hemodialysis, which increases plasma copper concentration [17,36], (ii) an effect of inflammatory tissue damage and pro-inflammatory status, both enhanced in KTR and positively correlated with copper status; and (iii) the influence of mycophenolate-based immunosuppressive therapies, that increase plasma copper when compared to other immunosuppressive medications [5,18]. Importantly, in our cohort of KTR, we found no differences in eGFR or proteinuria among tertiles of plasma copper concentration, and no association between plasma copper concentration and the risk of kidney graft failure. These results were expected since circulating copper is mainly bound to carrier proteins that do not easily cross the glomerular membrane. Therefore, the kidney copper load tends to remain low even if plasma copper concentration is increased [15,37].

In the overall KTR population, we found an association between plasma copper concentration and the risk of cardiovascular mortality, a major concern in the kidney post transplantation care since cardiovascular mortality is the major cause of death with a functioning graft [38]. Our findings are consistent with previous observational studies performed in the general population, which have also linked higher copper plasma concentration with cardiovascular disease and acute cardiovascular events [8,9,10,25,26,27]. Furthermore, in patients on hemodialysis, a positive association was also found between plasma copper concentration and the risk of all-cause mortality [39]. Multiple pathophysiological pathways that potentially explain the role of copper in enhancing cardiovascular disease in the general population are also applicable to KTR. Copper is a powerful promoter of LDL oxidation and modification to its atherogenic state [19,20,21]. Moreover, multiple experimental studies indicate that copper exposure can promote central inflammation [22,23]. In agreement, we found that in the overall KTR population, plasma copper concentration was positively associated with total and LDL cholesterol and hs-CRP concentration. KTR could also be more susceptible to copper-related damage because they have enhanced oxidative stress compared to the general population partly due to the immunosuppressive therapy [40,41,42], while copper has the capacity to participate in the formation of ROS and to propagate them [12,13,24]. Additionally, KTR possess a reduced concentration of copper-related defenses, namely the family of metallothionein proteins [43], which are intracellular metal-binding proteins that protect tissues from oxidative stress [44].

KTR on the third tertile of plasma copper concentration had, in general, a poorer cardiovascular profile, with e.g., higher BMI, higher systolic blood pressure, higher total and LDL cholesterol concentrations and higher diabetes prevalence. These differences may be related to the effect of copper on LDL oxidation and inflammation [19,20,21,22,23]. We accounted for the components of the poorer cardiovascular risk profile as potential confounders in the Cox regression analyses, finding that the association between plasma copper concentration and cardiovascular mortality remained materially unchanged both in the overall KTR population and among male KTR. These results agree with previous evidence in the general population, where investigators also described that the evaluated association between copper and cardiovascular disease was independent of other evaluated cardiovascular risk factors [8,9,10,25,26,27]. Noteworthy, in the overall KTR population, the association between plasma copper concentration and cardiovascular mortality was no longer significant after adjustment for hs-CRP. In male KTR, it did remain significant after this adjustment. Previous studies also reported that adjustment for inflammation parameters substantially decreases the association between plasma copper concentration and mortality risk [10]. These findings suggest there might be an overlap or interaction between the mechanisms by which copper and metabolic syndrome increase the risk of cardiovascular disease. Yet, the consistent persistence of this association after adjustment, both in the previously reported studies and among our study’s male KTR, suggests an independent mechanism.

There were considerable differences in copper status and effects among male and female KTR. First, consistent with multiple studies in the general population, we found that female KTR had higher plasma copper concentrations than male KTR [34,45,46]. Women are reported to have larger gastrointestinal absorption of copper compared to men [46]. Additionally, estrogen seems to increase plasma copper concentration further, as proposed by cross-sectional studies comparing women receiving estrogen-containing therapies with women who do not [34,47]. We also found that sex was a significant effect modifier of the association between plasma copper concentration and cardiovascular mortality. While male KTR displayed an association between cardiovascular mortality and plasma copper concentration, this association was absent in female KTR. The mechanisms by which copper increases cardiovascular risk have all been reported to have sex-differentiated effects. Copper-mediated LDL oxidation is known to be reduced by estrogens and therefore be lower among females [28]. Consistently, we found that among male but not female KTR, there was an association of plasma copper concentration with LDL and total cholesterol concentration. Moreover, females tend to have lower basal inflammation compared to males and a more robust anti-inflammatory response to inflammatory stressors because estrogen favors anti-inflammatory cytokines [48,49]. Finally, sex is also associated with differences in oxidative stress status and response. Previous studies have reported that the vascular concentration of ROS is lower in females than in males, and estrogen seems to have antioxidant properties since it can scavenge free radicals due to its phenolic hydroxyl group [50]. Of the referenced investigations connecting copper with cardiovascular disease in the general population and patients on hemodialysis, only one was composed of male participants [10], and the rest did not report the exploration of potential effect modification or stratified analyses by sex [8,9,27,39]. This phenomenon has been described in other fields, where reviews show that less than 10% of articles stratify results by sex [49]. Therefore, the effect modification by sex could have potentially been overlooked in previous studies. Another potential explanation of why we found this effect modification by sex while previous works did not, is that since the mechanism by which KTR are most susceptible to copper damage, namely oxidative stress, has been observed to be modified by sex [50], making the sex difference in this particular population with increased oxidative stress more evident.

The current study possesses several advantages: It comprises a large cohort of outpatient KTR extensively phenotyped, which has allowed adjustment for several potential confounders. The study did not have participant losses due to follow-up. Yet, it is a single-center study with a population almost entirely Caucasian, which calls for prudence when extrapolating our findings. Second, although we used the most commonly reported method for assessing copper status, we lacked other commonly used measurements, such as ceruloplasmin. We also lacked measurements of oxidized LDL and did not have information regarding plasma zinc concentration which could be a potential confounder of the reported association. However, previous studies have shown that copper’s association with cardiovascular disease is independent of zinc status [51,52]. Next, the study’s observational nature does not allow us to discern whether there is a causal link between copper and cardiovascular mortality or whether copper is a marker of higher risk. However, currently available evidence regarding copper biology supports that it has a pathophysiological link with cardiovascular disease [13,19,22,23,24].

## 5. Conclusions

In outpatient male KTR, a higher plasma copper concentration was independently associated with an increased risk of cardiovascular mortality. This study provides, for the first time, data that links copper status with cardiovascular outcomes in the post-kidney transplantation setting. This evidence also offers a rationale for considering potential sex differences in follow-up studies assessing copper effects in cardiovascular health and for further studies exploring copper-targeted strategies aimed at improving cardiovascular outcomes in KTR.

## Figures and Tables

**Figure 1 antioxidants-12-00454-f001:**
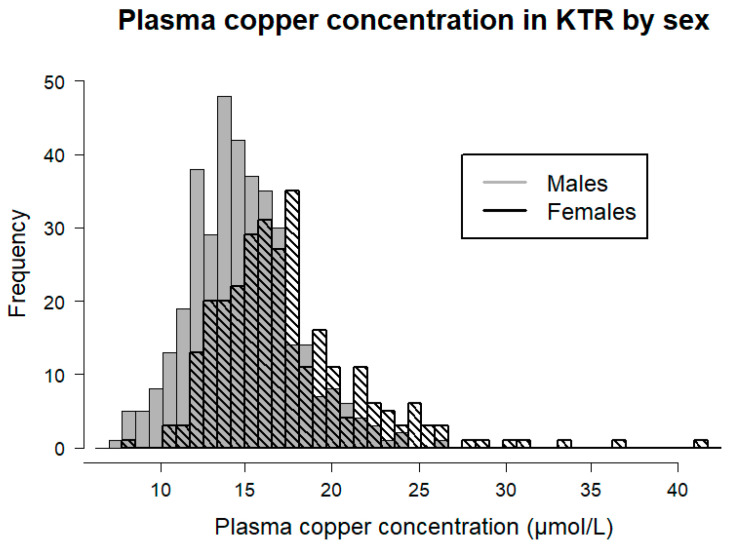
Sex-stratified plasma copper concentration in KTR. Median plasma copper concentration was 14.64 [12.75–16.37] µmol/L for male KTR and 16.68 [14.64–19.20] µmol/L for female KTR. Plasma copper concentration was significantly different among males and females according to the Mann-Witney U test (*p* < 0.001). KTR, kidney transplant recipients.

**Figure 2 antioxidants-12-00454-f002:**
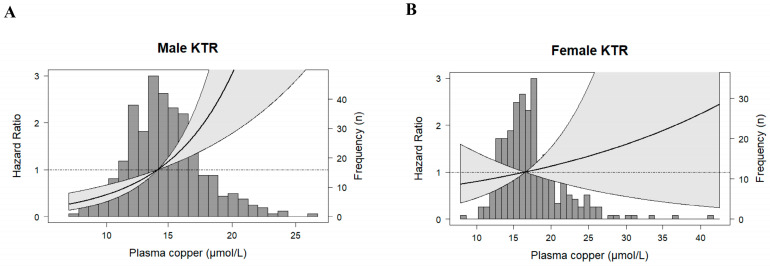
Sex-stratified analyses of the associations between plasma copper concentration and cardiovascular mortality in KTR. Male (**A**) and female (**B**) KTR data were fitted using a cubic spline-based Cox proportional-hazards regression model. The reference value was the median plasma copper concentration of 14.64 µmol/L for males (**A**), and 16.68 µmol/L for females (**B**). The solid line represents the hazard ratio. The grey area represents the 95% confidence interval. KTR, kidney transplant recipients.

**Figure 3 antioxidants-12-00454-f003:**
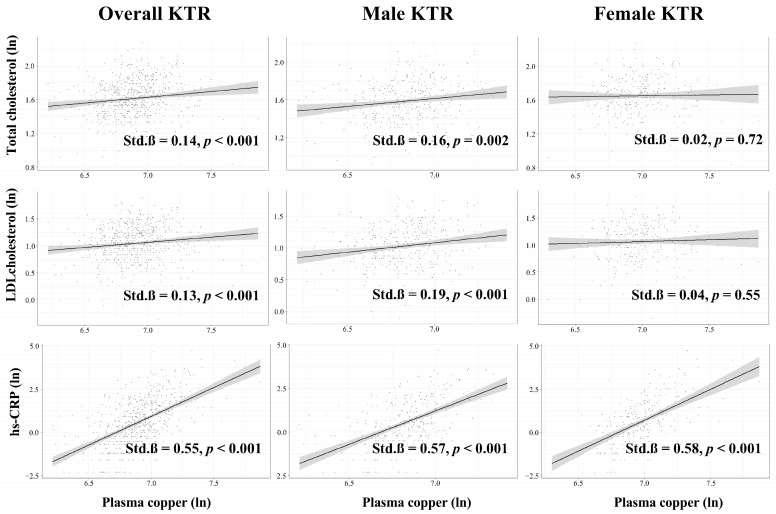
Association between plasma copper concentration and cardiovascular risk factors in KTR. Linear regression analyses among the overall KTR and among subgroups were performed. All variables were transformed to their natural logarithm. KTR, kidney transplant recipients; Std, standardized; LDL, low-density lipoprotein; hs-CRP, high-sensitivity C-reactive protein; ln, natural logarithm.

**Table 1 antioxidants-12-00454-t001:** Baseline characteristics of KTR.

	Overall KTR	Tertiles of Plasma Copper Concentration ^π^			*p* ^¥^
		Tertile 1	Tertile 2	Tertile 3	
*n*	660	220	220	220	—
Plasma copper concentration, µmol/L	15.42 (13.53–17.63)	12.59 (11.65–13.53)	15.42 (14.79–16.05)	18.88 (17.63–21.40)	—
**Demographics and body composition**					
Age, years	53 ± 13	52 ± 13	53 ± 13	54 ± 13	0.13
Sex (male), *n* (%)	370 (56)	161 (73)	130 (59)	79 (36)	<0.001
Caucasian ethnicity, *n* (%)	657 (99)	218 (99)	220 (100)	219 (99)	0.78
Body mass index, kg/m^2^	26.6 ± 4.8	26.0 ± 4.0	26.6 ± 4.7	27.4 ± 5.5	0.01
Waist circumference, cms ^a^	98 ± 15	97 ± 13	98 ± 14	100 ± 16	0.10
**Renal allograft function**					
eGFR, mL/min/1.73 m^2 b^	52 ± 20	53 ± 20	52 ± 19	51 ± 21	0.64
Proteinuria, *n* (%) ^b^	151 (23)	51 (23)	52 (24)	48 (22)	0.91
**Renal transplant**					
Preemptive transplantation, *n* (%)	103 (16)	40 (18)	36 (16)	27 (12)	0.22
Dialysis duration before transplantation, months ^c^	25 (9–50.0)	21 (5–49)	25 (10–50)	28 (11–48)	0.24
Living donor, *n* (%)	229 (35)	82 (37)	75 (34)	72 (33)	0.59
Donor age, years ^d^	43 ± 15	44 ± 15	42 ± 16	43 ± 15	0.37
Donor sex (male), *n* (%) ^e^	334 (52)	103 (47)	109 (51)	122 (57)	0.14
Time since transplantation, years	5.2 (1.8–12.1)	5.8 (2.3–12.1)	5.0 (1.8–12.3)	5.2 (1.5–10.8)	0.73
**Immunosuppressive therapy**					
Prednisolone use, *n* (%)	653 (99)	218 (99)	218 (99)	217 (99)	1.00
Calcineurin inhibitor use, *n* (%)	382 (58)	124 (56)	128 (58)	130 (59)	0.84
Proliferation inhibitor use, *n* (%)	549 (83)	181 (82)	183 (83)	185 (84)	0.88
Acute rejection treatment, *n* (%)	173 (26)	59 (27)	50 (23)	64 (29)	0.31
**Cardiovascular history**					
Cardiovascular disease, *n* (%)	158 (24)	43 (20)	51 (23)	64 (29)	0.06
Previous myocardial infarction, *n* (%)	32 (5)	7 (3)	11 (5)	14 (6)	0.30
Previous cerebrovascular event, *n* (%)	24 (4)	8 (4)	7 (3)	9 (4)	0.88
Previous vascular intervention, *n* (%)	65 (10)	17 (8)	28 (13)	20 (9))	0.19
Systolic blood pressure, mmHg	136 ± 17	136 ± 16	138 ± 18	133 ± 17	0.02
Diastolic blood pressure, mmHg	82 ± 11	83 ± 11	83 ± 12	81 ± 10	0.10
Antihypertensive use, *n* (%)	580 (88)	190 (86)	195 (89)	195 (89)	0.70
Total cholesterol, mmol/L	5.14 ± 1.14	4.89 ± 1.01	5.20 ± 1.20	5.33 ± 1.16	<0.001
Low–density lipoprotein–cholesterol, mmol/L ^b^	2.99 ± 0.94	2.79 ± 0.86	3.07 ± 0.96	3.11 ± 0.98	0.001
High–densitylipoprotein–cholesterol, mmol/L ^b^	1.39 ± 0.48	1.33 ± 0.47	1.38 ± 0.43	1.45 ± 0.53	0.05
Triglycerides, mmol/L	1.92 ± 1.02	1.85 ± 0.96	1.94 ± 1.11	1.98 ± 0.98	0.40
Statin use, *n* (%)	351 (53)	121 (55)	121 (55)	109 (50)	0.42
Diabetes, *n* (%)	156 (24)	40 (18)	47 (21)	69 (31)	0.003
Hemoglobin, mmol/L ^b^	8.16 (1.05)	8.33 (1.03)	8.22 (1.05)	7.93 (1.03)	<0.001
**Inflammation and oxidative stress**					
Leukocyte count, 10^9^/L ^b^	7.70 (6.30–9.50)	7.80 (6.45–9.30)	7.50 (6.10–9.50)	7.70 (6.10–9.70)	0.85
High–sensitivity C–reactive protein, nmol/L ^f^	15.24 (7.43–43.81)	6.66 (2.86–13.33)	15.71 (8.57–29.71)	43.81 (16.19–97.14)	<0.001
Plasma malondialdehyde, µmol/L ^g^	2.53 (1.91–3.78)	2.57 (2.04–3.67)	2.47 (1.84–4.05)	2.55 (1.97–3.79)	0.75
**Lifestyle**					
Smoking behavior, *n* (%) ^h^					0.53
Current	78 (13)	25 (12)	32 (15)	21 (10)	
Previous	285 (46)	94 (45)	90 (43)	101 (49)	
Never	263 (42)	89 (43)	90 (43)	84 (41)	
Alcohol intake >30 g/day, *n* (%) ^i^	28 (5)	8 (4)	10 (5)	10 (5)	0.85
SQUASH score, minutes/week × intensity	5070 (2040–8048)	5610 (1800–7660)	5520 (3145–8820)	4380 (1432–6878)	0.001

^π^ Tertile 1: <0.89 mg/L, tertile 2: 0.89–1.07 mg/L, tertile 3: >1.07 mg/L. ^¥^ Differences were tested using ANOVA for continuous variables with normal distribution, Kruskal–Wallis test for continuous variables with non–normal distribution and using χ^2^ test for categorical variables. Data available in ^a^ 638, ^b^ 659, ^c^ 642, ^d^ 645, ^e^ 635, ^f^ 624, ^g^ 655, ^h^ 621 and ^i^ 595 KTR. KTR, kidney transplant recipients; eGFR, estimated glomerular filtration rate, SQUASH, Short QUestionnaire to ASsess Health–enhancing physical activity.

**Table 2 antioxidants-12-00454-t002:** Association between plasma copper concentration and cardiovascular mortality in KTR.

Cardiovascular Mortality	Plasma Copper Concentration (Ln, per 1-SD Increment)		
	HR	95% CI	*p*
Crude	1.37	1.07–1.77	0.01
Model 1	1.48	1.11–1.98	0.01
Model 2	1.47	1.10–1.95	0.01
Model 3	1.49	1.11–1.99	0.01
Model 4	1.40	1.05–1.88	0.03
Model 5	1.40	1.06–1.84	0.02
Model 6	1.06	0.73–1.54	0.75

In total, 141 (21%) KTR died, 54 (38%) due to cardiovascular causes. Model 1: adjustment for age, sex and BMI. Model 2: model 1 + eGFR, time since transplantation and dialysis time before transplantation. Model 3: model 1 + calcineurin inhibitor use, proliferation inhibitor use. Model 4: model 1 + SBP, history of previous cardiovascular disease and hemoglobin. Model 5: model 1 + total cholesterol, LDL cholesterol, diabetes history. Model 6: model 1 + SQUASH score + hs-CRP.

**Table 3 antioxidants-12-00454-t003:** Association between plasma copper concentration and cardiovascular mortality in male KTR.

Cardiovascular Mortality	Plasma Copper Concentration (Ln, Per 1-SD Increment)		
	HR	95% CI	*p*
Crude	2.09	1.42–3.07	<0.001
Model 1	1.84	1.24–2.73	0.003
Model 2	1.75	1.16–2.61	0.01
Model 3	1.88	1.27–2.80	0.002
Model 4	1.72	1.16–2.55	0.01
Model 5	1.80	1.21–2.69	0.004
Model 6	1.55	1.01–2.37	0.04

In total, 83 (22%) male KTR died, 35 (42%) due to cardiovascular causes. Model 1: adjustment for age and BMI. Model 2: model 1 + eGFR, time since transplantation and dialysis time before transplantation. Model 3: model 1 + calcineurin inhibitor use, proliferation inhibitor use. Model 4: model 1 + SBP, history of previous cardiovascular disease and hemoglobin. Model 5: model 1 + total cholesterol, LDL cholesterol, diabetes history. Model 6: model 1 + SQUASH score + hs-CRP.

## Data Availability

The data presented in this study are available on request from the corresponding author. The data are not publicly available due to patient privacy protection policies.

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
