# Peer review of "Plasma Copper Concentration Is Associated with Cardiovascular Mortality in Male Kidney Transplant Recipients"

_antioxidants, 2023, doi:10.3390/antiox12020454_

Round 1

Reviewer 1 Report

The authors investigated whether, in KTR, post-transplantation copper status is associated with the risk of cardiovascular mortality and potential effect modification by sex. The theme is very relevant and important for the research area. The work is well-written and well-organized. The research is quite relevant and the results are promising. I only make a suggestion to clarify whether higher plasma copper concentrations could have affected the function of transplanted kidneys.

Author Response

Manuscript ID: antioxidants-2175844

Detailed, itemized response to the comments of Reviewer #1.

Reviewer #1

The authors investigated whether, in KTR, post-transplantation copper status is associated with the risk of cardiovascular mortality and potential effect modification by sex. The theme is very relevant and important for the research area. The work is well-written and well-organized. The research is quite relevant and the results are promising. I only make a suggestion to clarify whether higher plasma copper concentrations could have affected the function of transplanted kidneys.

      Response: We thank the Reviewer for the kind appraisal of our work and the comment. The vast majority of copper circulates in plasma bound to carrier proteins, namely ceruloplasmin and albumin [1]. Therefore, copper load in the kidney tends to remain low even when plasma concentrations are increased because the carrier proteins do not easily cross the glomerular membrane [2]. In our cohort of KTR, there was no difference among tertiles of plasma copper concentration in the estimated glomerular filtration rate or the presence of proteinuria. This suggests that the graft condition was not affected by the plasma copper concentration. Furthermore, we evaluated the association of plasma copper concentration with graft failure. Of note, there was no significant  association between plasma cooper copper concentration and the risk of graft failure in the overall and male KTR (HR 0.85, 95% CI 0.68–1.07 per 1-SD increment, P=0.16 and HR 0.86, 95% CI 0.62-1.18 per 1-SD increment, P=0.35; respectively). We also accounted for graft function as a potential confounder in the association of plasma copper concentration and cardiovascular mortality in KTR (model 2). Our findings on the association between copper and risk of cardiovascular mortality in the overall and male KTR remained materially unchanged after this adjustment (HR 1.47, 95% CI 1.10–1.95 per 1-SD increment, P=0.01 and HR 1.75, 95% CI 1.16-2.61 per 1-SD increment, P=0.01; respectively).

To accommodate the Reviewer's comment, we added the results regarding graft failure to the results section and the supplemental material (page 8, line 250-254; Table S4. Association of plasma copper concentration with all-cause mortality, non-cardiovascular mortality and graft failure in KTR). We also added a comment on this results on the Discussion section of the revised version of the manuscript, which now reads: "Importantly, in our cohort of KTR, we found no differences in eGFR or proteinuria among tertiles of plasma copper concentration, and no association between plasma copper concentration and the risk of kidney graft failure. There results were expected since circulating copper is mainly bound to carrier proteins that do not easily cross the glomerular membrane. Therefore the kidney copper load tends to remain low even if plasma copper concentration is increased [1,2]” (page 9, line 281-286).

REFERENCES

  1. Gaetke, L.M.; Chow-Johnson, H.S.; Chow, C.K. Copper: Toxicological Relevance and Mechanisms. Arch Toxicol 2014, 88, 1929–1938, doi:10.1007/s00204-014-1355-y.
  2. Ito, S.; Fujita, H.; Narita, T.; Yaginuma, T.; Kawarada, Y.; Kawagoe, M.; Sugiyama, T. Urinary Copper Excretion in Type 2 Diabetic Patients with Nephropathy. Nephron 2001, 88, 307–312, doi:10.1159/000046013.

Reviewer 2 Report

I considered the manuscript entitled “Plasma copper concentration is associated with cardiovascular mortality in male kidney transplant recipients” by Manuela Yepes-Calderón, et al, that is intended to be published in Antioxidants journal.

In this computational study, at last it is not clear if copper concentration is just a factor of coincidence or causality for the cardiovascular mortality. The study appears well performed but it is a replica of what happens in the general population in a specific population. Authors need to enrich the discussion to introduce additional interest for the readers. This event has already been described in the dialysis population.

Authors suggest several explanations for the found  association between high copper concentrations and CV mortality: (i) a remanent effect from hemodialysis, which increases plasma copper concentration, (ii) an effect of inflammatory tissue damage and pro-inflammatory status, both enhanced in KTR and positively correlated with copper status; and (iii) the influence of mycophenolate-based immunosuppressive therapies, that increase plasma copper when compared to other immunosuppressive medications. These are plausible explanations, but they occur similarly in women and men. Then, how can you explain the sex differences? And, these explanations fit with the KTR group but the described association also occurs in the general population where other mechanisms may occur…

Following your discussion concerning the correlation of copper concentration and inflammation, it appears that patients with cardiovascular mortality apart from higher copper concentration also present higher number of cardiovascular factors. For instance, there is a harder history of diabetes in those patients with the higher copper concentrations and cardiovascular mortality. It will not be that the concentration of copper is framed in the mechanisms of metabolic syndrome?

You analyze current smoking, but what about the past history of smoking?? What about previous peripheral artery disease or cardiovascular events? Which was the duration of the dialysis of these patients, especially for those who died?

What is your opinion concerning the discrepancies in the sex differences in your studied KTR population and the lack of sex differences in other studies?

Reviewer 3 Report

In this manuscript, authors demonstrated that high plasma copper levels, in conjugation with elevated high-sensitivity C-reactive protein, were significantly associated with increased risk of cardiovascular death in male kidney transplant recipients in a single center cohort study. The study was well designed, and the subject of study seems to be interesting for many readers. There are some comments and questions to the manuscript, as follows.

1. The timing when plasma copper levels were determined by inductively couped plasma mass spectrometry appears to be unclear. Authors have to specifically describe the timing of copper measurement in the method section.

2. Data is derived from relatively large cohort of kidney transplant recipients. However, follow up period in this cohort is unclear.

3. Copper is generally related to inflammation with mild elevation of C-reactive protein as well as LDL cholesterol levels as authors suggested in Discussion. However, it can affect other metabolic factors. Since it has been reported that lower zinc levels are associated with cardiovascular death, the effect of higher copper on plasma zinc level may be a confounding factor. In addition, it is better to present hemoglobin levels and if needed, include it in the analyses.

Author Response

Manuscript ID: antioxidants-2175844

Detailed, itemized response to the comments of Reviewer #3.

Reviewer #3

In this manuscript, authors demonstrated that high plasma copper levels, in conjugation with elevated high-sensitivity C-reactive protein, were significantly associated with increased risk of cardiovascular death in male kidney transplant recipients in a single center cohort study. The study was well designed, and the subject of study seems to be interesting for many readers. There are some comments and questions to the manuscript, as follows.

Response: We thank the Reviewer for the kind appraisal of our work. Please further see our reactions to the specific comments listed below.

Specific comments

Comment #1. The timing when plasma copper levels were determined by inductively couped plasma mass spectrometry appears to be unclear. Authors have to specifically describe the timing of copper measurement in the method section.

Response: We thank the Reviewer for the comment. KTR provided the plasma samples from which copper plasma concentration was measured at the first medical visit after enrollment, at least one-year post-transplantation, and at a median of 5.2 (IQR 1.8-12.1) years after transplantation. To accommodate the comment of the Reviewer, we now clarify this information in the "Materials and Methods" section of the revised version of the manuscript under the subheading "Laboratory measurements, calculations, and definitions", which now reads as follows: "During the first visit to the outpatient clinic, fasting blood samples were drawn from all participants at a median of 5.2 (IQR 1.8–12.1) years after transplantation and aliquots of plasma generated from these samples were stored frozen at -80°C. Plasma copper was afterwards measured by inductively coupled plasma mass spectrometry (ICPMS) ICAP Q instrument (Thermo Fisher Scientific, Waltham, MA, USA) at Synalb (Jena, Germany)" (page 2, line 85-90). Since the time of taking samples for plasma copper assessment was variable among KTR, we accounted for this as a potential confounder in our Cox regression analyses (model 2). Of note, our findings on the association between copper and risk of cardiovascular mortality in the overall KTR population and in male KTR remained materially unchanged after this adjustment (hazard ratio (HR 1.46, 95% CI 1.10–1.95 per 1-SD increment, P=0.01 and HR 1.81, 95% CI 1.23–2.66 per 1-SD increment, P<0.001; respectively).

Comment #2. Data is derived from relatively large cohort of kidney transplant recipients. However, follow up period in this cohort is unclear.

Response: We thank the Reviewer for the comment. After inclusion, KTR were followed for a median of 5.4 (IQR 4.8–6.1) years. To accommodate the comment of the Reviewer, we now clarify this information in the "Materials and Methods" section of the revised version of the manuscript under the subheading "Data collection", which reads as follows: "Endpoints were recorded until September 2015, resulting in a median follow-up of 5.4 (IQR 4.8–6.1) years after inclusion. We contacted general practitioners or referring nephrologists in cases where the status of a patient was unknown. No participants were lost to follow-up". This information can also be found under the sub-heading "Longitudinal analyses" of the "Results" section (page 2, line 79-83 and page 6, line 182).

Comment #3. Copper is generally related to inflammation with mild elevation of C-reactive protein as well as LDL cholesterol levels as authors suggested in Discussion. However, it can affect other metabolic factors. Since it has been reported that lower zinc levels are associated with cardiovascular death, the effect of higher copper on plasma zinc level may be a confounding factor. In addition, it is better to present hemoglobin levels and if needed, include it in the analyses.

Response: We thank the Reviewer for the comment. Unfortunately, the relationship between plasma copper and zinc concentration could not be explored because zinc measurements were unavailable in this cohort. To accommodate the comment of the Reviewer, we have added this limitation to the "Discussion" section of the revised version of the manuscript, which now reads as follows: "[…] we did not have information regarding plasma zinc concentration which could be a potential confounder of the reported association. However, previous studies have shown that copper's association with cardiovascular disease is independent of zinc status [1,2]” (page 10, line 364-367).

Following the other part of the comment of the Reviewer, we have added the information regarding hemoglobin concentration to Table 1 ("Baseline characteristics of KTR"), Table S1 ("Baseline characteristics of male KTR"), and Table S2 ("Baseline characteristics of female KTR"). In the overall KTR population, hemoglobin concentration was higher among patients in the first tertile of plasma copper concentration (P<0.001). Therefore, we have also accounted for this potential confounder in our Cox regression analyses (model 4). Of note, our findings on the association between copper and risk of cardiovascular mortality in the overall KTR population and in male KTR separately remained materially unchanged after this adjustment (HR 1.40, 95% CI 1.05–1.88 per 1-SD increment, P=0.03 and HR 1.72, 95% CI 1.16-2.55 per 1-SD increment, P=0.01; respectively).

REFERENCES

  1. Shokrzadeh, M.; Ghaemian, A.; Salehifar, E.; Aliakbari, S.; Saravi, S.S.S.; Ebrahimi, P. Serum Zinc and Copper Levels in Ischemic Cardiomyopathy. Biol Trace Elem Res 2009, 127, 116–123, doi:10.1007/s12011-008-8237-1.
  2. Kunutsor, S.K.; Voutilainen, A.; Kurl, S.; Laukkanen, J.A. Serum Copper-to-Zinc Ratio Is Associated with Heart Failure and Improves Risk Prediction in Middle-Aged and Older Caucasian Men: A Prospective Study. Nutrition, Metabolism and Cardiovascular Diseases 2022, 32, 1924–1935, doi:10.1016/j.numecd.2022.05.005.

Round 2

Reviewer 2 Report

no further comments

Author Response

We thank the reviewer for his/her time and effort. 

Reviewer 3 Report

Authors have addressed most of the reviewer's concerns in the revised manuscript. However, authors should clearly describe that plasma samples were collected at first visit after study inclusion in the method section. The relationship between transplantation and inclusion appears to be uncertain.

Author Response

Response: We thank the Reviewer for the comment. To accommodate the comment of the Reviewer, we now clarify this information in the "Materials and Methods" section of the revised version of the manuscript under the subheading "Laboratory measurements, calculations, and definitions", which now reads as follows: "During the first medical visit after study inclusion, at a median of 5.2 (IQR 1.8–12.1) years after transplantation, all participants provided fasting blood samples. Aliquots of plasma generated from these samples were stored frozen at -80°C and plasma copper was afterwards measured by inductively coupled plasma mass spectrometry (ICPMS) ICAP Q instrument (Thermo Fisher Scientific, Waltham, MA, USA) at Synalb (Jena, Germany)" (page 2, line 85-89).